# Southern Ethiopian skilled birth attendant variations and maternal mortality: A multilevel study of a population-based cross-sectional household survey

Aschenaki Zerihun Kea [1,2]*, Bernt Lindtjørn[1,2], Achamyelesh Gebretsadik Tekle[1], Sven Gudmund Hinderaker[2]

1 School of Public Health, College of Medicine and Health Sciences, Hawassa University, Hawassa, Ethiopia, 2 Centre for International Health, University of Bergen, Bergen, Norway

* aschenakizer@yahoo.com, aschekea@gmail.com

## Abstract

Studies examining skilled birth attendants (SBA) use and its correlation with maternal mortality at lower administrative levels are scarce. This study assessed the coverage and variations of SBA, the physical accessibility of health facilities for SBA, and the association of SBA with maternal mortality. A cross-sectional study using a population-based household survey was conducted in six Sidama National Regional State, southern Ethiopia districts, from July 2019 to May 2020. Women who had given birth in the past two years before the study were included. Stata 15 and ArcGIS 10.4.1 were used for data analysis. A multilevel logistic regression analysis was conducted to assess the effect of the sampling units and identify factors independently associated with SBA. The association between SBA and maternal mortality was examined using maternal mortality household survey data. A total of 3191 women who had given birth in the past two years and resided in 8880 households sampled for the associated maternal mortality household survey were interviewed. The coverage of SBA was 46.7%, with high variations in the districts. Thirty percent of SBA use was accounted for by the differences among the districts. One-third of the women travel more than two hours on foot to access the nearest hospital. Districts with low coverage of SBA and located far away from the regional referral centre had high maternal mortality. Education of the mother, occupation of the husband, pregnancy-related complications, use of antenatal care, parity, and distance to the nearest hospital and health centre were associated with the use of SBA. The coverage of SBA in the Sidama Region was low, with high variations in the districts. Low SBA use was associated with high maternal mortality. Due attention should be given to districts with low coverage of SBA and those located far away from the referral centre. Access to hospitals has to improve. All women should be encouraged to get antenatal care services.

**Data Availability Statement:** The relevant data are available at Open Science framework: DOI: https://doi.org/10.17605/OSF.IO/9G6PW.

**Funding:** This study was funded by Grand Challenges IDRC Canada through the Liverpool School of Tropical Medicine and REACH Ethiopia, a non-profit organization. The funders had no role in study design, data collection and analysis, decision to publish, or preparation of the manuscript.

**Competing interests:** The authors have declared that no competing interests exist.

## Introduction

The time around labour, childbirth, and the first week post-delivery holds a higher risk of death for mothers and new-borns [1]. Hence, every mother should be assisted by a skilled birth attendant (SBA) during this critical time. SBA is defined as a competent maternal and new-born health professional, trained and regulated to national and international standards [2]. It includes doctors, midwives, and nurses who perform all signal functions of emergency maternal and new-born care to optimise the health and well-being of women and new-borns [2].

Globally, a skilled attendant at birth has been considered one of the key interventions to reduce maternal mortality and an indicator used to monitor the progress of maternal mortality reduction [3]. Studies conducted using the data of countries from international databases and systematic reviews demonstrated that there existed an inverse relationship between SBA and maternal mortality: high coverage of SBA was associated with low maternal mortality [4–6]. However, the effect of SBA on maternal mortality is not well studied at the subnational and district levels using primary data.

In low and middle-income countries (LMICs), access to maternal health services is poor [7], and the coverage of skilled birth attendance is low [8–10]. As a result, many births are attended by unskilled birth attendants, like traditional birth attendants, neighbours, mothers-in-law, and husbands; some even take place without any assistance [11, 12]. Besides poor access and low coverage of SBA in LMICs, the existing health care is not uniformly accessible to all segments of the population [13–17]. Reaching every woman by minimising inequalities in access and improving the coverage of care is one of the strategic objectives to end preventable maternal mortality [18].

The utilisation of SBA services can be affected by several factors related to maternal characteristics such as the age of the mother [19], educational status [10], use of antenatal care (ANC) [10], parity [19], pregnancy-related complications [12], household characteristics such as the level of husband education [9], occupation of the husband [20], wealth status [9, 19], and distance or travel time required to access a health facility [21].

Like other LMICs, Ethiopia has an unacceptably high maternal mortality [22, 23]. Despite an increasing trend in the coverage of skilled birth attendance since 2000 [24], coverage and access to SBA are still low in the country [10, 16]. In Ethiopia, there is a dearth of studies conducted to assess the utilisation of skilled birth attendants [10, 25, 26]. However, this study tries to answer the following questions: what is the coverage and disparities of SBA in the Sidama Region and the districts, how physically accessible the health centres and hospitals are to pregnant women, how they are utilized by the women during labour and childbirth, and does SBA affect the magnitude of maternal mortality in districts of Sidama Region.

We conducted this study as part of a larger study that measured maternal mortality in Sidama National Regional State, employing a population-based household survey with a five-year recall of pregnancy and birth outcomes and the sisterhood method [11, 27]. We found that the maternal mortality ratio (MMR) was 419 per 100,000 live births (LB) by five-year recall and 623 per 100,000 LB by the sisterhood method, with large variations by district [11, 27]. The studies also identified that districts far from the regional capital, with poor infrastructure and inadequate skilled health personnel, had a higher rate of maternal mortality [11, 27].

Hence, we carried out this study with the following specific objectives: 1) To assess the SBA coverage in Sidama National Regional State and its variations within the districts of the region; 2) To identify factors associated with SBA; 3) To measure the accessibility of health centres and hospitals for utilization of SBA services employing geographic information system; 4) To examine the effect of SBA use on the reduction of maternal mortality.

## Materials and methods

### Study design, setting, and period

A population-based cross-sectional household survey was conducted in six districts (Aleta Chuko, Aleta Wondo, Aroresa, Daela, Hawassa Zuriya, and Wondogenet) of Sidama National Regional State, southern Ethiopia, from July 2019 to May 2020. Sidama National Regional State is one of the 12 regional states in the country, and its capital, Hawassa, is located 273 km south of Addis Ababa. In 2020, the region had a total population of 4.3 million people [28]. Ninety-two percent of the population lives in rural areas, and agriculture is the main livelihood of the population [29].

Administratively, the Sidama Region is divided into 30 rural districts (*woredas*), 6 town administrations, and 536 rural *kebeles* (the smallest administrative structure with an average population of 5000 people). Under the kebele is a local structure known as *limatbudin* (a local administrative unit consisting of 40–50 neighbouring households on average).

In Sidama National Regional State, there are 18 hospitals (13 primary, 4 general, and 1 tertiary), 137 health centres, and 553 health posts owned and operated by the government [30]. In addition, the region has 4 hospitals (1 general and 3 primary), 21 specialty and higher clinics, 131 medium clinics, and 79 primary clinics operated by private owners.

### Study population and sampling techniques

Women in the reproductive age group of 15–49 years in Sidama National Regional State who had given birth in the past two years preceding the survey were the source population. Women of the same age range who resided in sampled households and had given birth two years before the survey were the study population. In the period between 1 July 2019 and 31 May 2020, a total of 3191 women who had given birth in the past two years before the study were recruited from the sampled households in Sidama Region.

The sampling techniques of the study have been described elsewhere [11]. Below, we briefly describe the summary of the sampling technique. We employed multistage cluster sampling techniques. Firstly, we randomly selected six districts out of the 30 districts in the region. Secondly, we selected 40 kebeles (the smallest administrative structure with an average population of 5000 people) from the six districts, proportional to the size of the kebeles. Thirdly, from each kebele, we sampled six *limatbudin* (administrative units organised by 40–50 neighbouring households), and finally, we selected 37 households from each *limatbudin*. In the sampled households, we interviewed women who had given birth in the past two years before the survey. Women who had given birth more than two years before the survey were excluded from the study. We also excluded women who relocated to the study area in the past two years prior to the study. If a woman had given birth twice in the past two years before the survey, we chose the most recent birth experience for the study.

### Variables

The study's main outcome measure was the use of a skilled attendant during childbirth, which was derived from the question, "Who assisted the delivery of your last birth?" This gives a dichotomized response that categorises the study participants into two groups: women who used skilled birth attendants and those who used unskilled birth attendants. Operationally, we defined a woman as having used a skilled birth attendant when she gave birth with the assistance of skilled health personnel such as nurses, midwives, health officers, and doctors [2]. The delivery was conducted at government health facilities: health centres and hospitals, faith-based clinics, private clinics, or hospitals. In contrast, we defined women who used unskilled

birth attendants as those who gave birth with the assistance of unskilled birth attendants like health extension workers (HEWs), TBAs (including both trained and untrained TBAs), mothers-in-law, husbands, and neighbours [15, 31].

According to the Ethiopian health system, health centres are responsible for basic emergency obstetric and new-born care (BEmONC), including delivery services. Hospitals provide comprehensive emergency obstetric care (CEmONC), including all BEmONC services, obstetric surgery, and blood transfusion [32]. Health posts, run by the HEWs, provide disease prevention and health promotion activities and some basic curative services [33]. As the government of Ethiopia promotes SBA, the HEWs do not provide delivery services as part of their routine duties except to assist mothers who come to their knowledge during an emergency and facilitate referral to the catchment health centre or primary hospital [33].

The explanatory variables in this study include background characteristics of women like age, education, history of parity, ANC visits, pregnancy-related complications, family or household characteristics such as the level of education of the husband, occupation of the head of household, socioeconomic status (wealth index) of the household, and distance from the residential location to the nearest health centre and the nearest hospital.

## Data source and measurement

An interviewer-administered questionnaire was used to collect the data from women who were of childbearing age, 15–49 years old, and who had given birth in the past two years before the study. The data was collected by diploma-level teachers recruited from each *kebele* who were familiar with the culture and language of the study area.

During the household visit, the data collectors recorded the geographic coordinates (latitude and longitude) of residential locations, the nearest health centres, and hospitals using the Garmin ETrex 10 handheld global positioning system (GPS).

The age of a woman was collected and registered in a completed year. The educational level of the woman and husband was registered as "completed grade" and further categorised as "no formal education,"grade 1–4," "grade 5–8," "grade 9–12," and "high school or above." The occupation of the head of the household was identified as the main source of income and registered as a farmer, trader, government employee, and others. The history of pregnancy-related complications was assessed using questions eliciting whether a woman had pregnancy-related complications during pregnancy or childbirth.

The household's wealth status was estimated by creating a wealth index using household assets and facility variables. We used principal component analysis (PCA) to create a household wealth index [34]. The wealth index initially had five centiles: poorest, poor, middle, rich and richest. Later, we classified the poorest, poor and middle as "poor" and the rich and richest as "rich" for analysis. The variables used for wealth index creation have been described elsewhere [11].

The distance from the residential location to the nearest health centre and hospital was calculated by a straight line or Euclidean distance [35, 36] employing the "Near Analysis" function in ArcGIS 10.4.1.

## Data quality control

The questionnaire for the study was prepared after reviewing the relevant literature. We initially prepared the questionnaire in English, translated it into the local language (*Sidaamu Afoo*), and had it back-translated to English by another individual to check its consistency. Before data collection, the questionnaire was pretested in one district not included in the study.

The principal investigator trained the data collectors and supervisors for three days. The training included discussion on the content and aim of each question, interview techniques, role plays, how to operate, read, and record geographical coordinates using GPS (Garmin ETREX 10), and a field test.

Two public health officers recruited from each district who were familiar with the culture and language of the study setting supervised the data collectors. The supervisors followed the data collectors and checked the consistency and completeness of the questionnaire daily. When eligible respondents were absent during the initial visit, the data collectors revisited the households the next day.

The data was double entered using EpiData software (EpiData Association 2000–2021, Denmark). The consistency of the two entries was checked, and discrepancies were validated from the hard copies of the questionnaire. To maintain the quality of spatial data, the coordinates of residential locations, the nearest health centres, and the nearest hospitals were recorded on a GPS device and paper and double-checked for consistency.

To confirm the place of delivery reported by mothers, we asked different verification and probing questions: asked a woman to mention the name of the health facility she had given birth to, whether a birth certificate was provided or not during childbirth, whether the baby was vaccinated, and whether immediate post-natal counselling was given to the mother.

## Sample size determination

The estimated sample size for this study was determined based on the following assumptions: 50% prevalence of skilled delivery [37], a design effect of 2 (as the study employed the multi-stage cluster sampling method), and precision +/-5%. Hence, the sample size calculated was 768 women who gave birth in the past two years prior to the survey. Adding 10% non-response, the final sample size needed was 845 women.

This study is part of a larger maternal mortality study, and the sample of households needed for the larger study was 8880 households, with the corresponding number of mothers [11]. We assumed that surveying the 8880 households would be sufficient to get the desired sample of women who had given birth in the past two years before the study, which enables us to estimate the coverage of SBA.

## Data analysis

We used Stata 15 (Stata Corp., College Station, TX) for data analysis. Frequency analysis like mean (SD), proportions, and percentages were computed to describe the outcome and explanatory variables. A bivariate analysis using the Chi-square test was conducted to identify factors associated with the outcome variable; SBA. To account for the hierarchical nature of the data, where individual mothers are nested within a household, households are nested in *limatbudin*, *limatbudins* are nested in kebeles, and kebeles are nested in districts, a multilevel mixed-effect logistic regression model (MLLR) was employed.

The MLLR analysis aimed to understand the extent of variations in SBA accounted for by the three sampling units (*limatbudin*, kebele, and district) and assess factors independently associated with the outcome variable. Variables with a p-value <0.2 in the bivariate analysis were considered for MLLR analysis [12]. Four models were fitted. The first model was the null model, which had no explanatory variables. The null model was used to assess the random effect of the sampling units. The second model contained individual variables, the third model adjusted for household-level factors, and the fourth included individual and household-level variables. Intra-cluster correlation (ICC) was used to assess the random effect. Model fitness was assessed using Akaike's Information Criterion (AIC).

To assess the strength of the association between explanatory variables and the outcome variable, crude and adjusted odds ratios with a 95% confidence interval (CI) were computed. Associations with a p-value <0.05 in the MLLR model were considered significant. A stratified analysis was conducted to assess confounding and effect modification using the estimates of stratum-specific, crude, and adjusted odds ratios.

We also assessed the association between the coverage of SBA and the maternal mortality ratio (MMR). The MMR data was obtained from a maternal mortality household survey conducted in the study area where the current study was conducted as part of the study [11].

We assumed women residing in locations within 10 km of the nearest health centre or the nearest hospital have good physical accessibility to SBA services [38]. In contrast, women who live in areas outside 10 km of the nearest health centre or the nearest hospital are assumed to have poor physical accessibility to SBA services [38]. ArcGIS 10.4.1 (ESRI, Redlands, CA, USA) software measured the distance between residential locations and the nearest hospital and health centre. We assumed 2 hours of walking time are required to travel 10 km [39].

### Ethical approval

We obtained ethical approval for this study from the institutional review board of Hawassa University College of Medicine and Health Sciences (IRB/015/11) and the Regional Ethical Committee of Western Norway (2018/2389/REK Vest). We got support letters from the Sidama National Regional State Health Bureau (formerly known as the Sidama Zone Health Department), the district health offices, and the kebele administrations. Informed (thumbprint and signed) consent was obtained from the respondents. To maintain the anonymity of the study participants, we did not collect any personal identifying information during data collection. In this study, we did not find minors (women below the age of 18 years). Hence, we did not require consent from parents or guardians.

### Inclusivity in global research

Additional information regarding the ethical, cultural, and scientific considerations specific to inclusivity in global research is included in S1 Checklist.

## Result

### Sociodemographic and obstetric backgrounds of study participants

Table 1 describes the participant women's sociodemographic and obstetric backgrounds and household characteristics. 3191 (100%) women of childbearing age (15–49 years) were interviewed. The mean age of the women was 25.2 years, and 145 (4.5%) were under the age of 20 years.

Twenty-two per cent, 698 (21.9%) women, had no formal education, 790 (24.8%) did not have antenatal care (ANC), and 2401 (75.2%) had at least one antenatal visit. Of those with ANC, 853 (35.5%) had four or more ANC visits. Mothers who reported at least one pregnancy-related complication during pregnancy or childbirth were 1246 (39%). The husbands of women without formal education were 420 (13.2%). Three-fifths of the women, 1932 (60.5%), were from poor households, and 2276 (71.3%), the heads of the households, were engaged in farming activities.

### Place of delivery and assistance during delivery

Table 2 presents the place and assistance for delivery. We found that 46.9% of mothers delivered at health facilities and 53.1% delivered at home. Among mothers who had given birth at a

**Table 1. Background characterstics of study women and households, Sidama National Regional State, 2020.**

| Variable | Number | Percent |
| --- | --- | --- |
| **Age of mother:** | | |
| 15–19 | 145 | 4.5 |
| 20–24 | 1146 | 35.9 |
| 25–29 | 1147 | 35.9 |
| 30–34 | 546 | 17.1 |
| 35+ | 207 | 6.8 |
| **Education of mother** | | |
| No formal education | 698 | 21.9 |
| 1–4 Grade | 1004 | 31.5 |
| 5–8 Grade | 1102 | 34.5 |
| 9–12 Grade (High School) | 316 | 9.9 |
| Above high school | 71 | 2.2 |
| **Had ANC** | | |
| No | 790 | 24.8 |
| Yes | 2401 | 75.2 |
| **Number of ANC visits** | | |
| Less than 4 ANC visits | 1548 | 64.5 |
| Four or more ANC visits | 853 | 35.5 |
| **Had at least one pregnancy related complication** | | |
| No | 1945 | 61.0 |
| Yes | 1246 | 39.0 |
| **Education of husband** | | |
| No formal education | 420 | 13.2 |
| 1–4 Grade | 730 | 22.9 |
| 5–8 Grade | 1393 | 43.7 |
| 9–12 Grade (High School) | 461 | 14.4 |
| Above high school | 187 | 5.9 |
| **Wealth index of household** | | |
| Poor | 1932 | 60.5 |
| Rich | 1259 | 39.5 |
| **Occupation of household head** | | |
| Farming | 2276 | 71.3 |
| Non-farming (trading, gov't employee etc.) | 915 | 28.7 |
| **Distance to the nearest hospital** | | |
| Outside 10 km | 1052 | 33.0 |
| Below 10 km | 2139 | 67.0 |
| **Distance to the nearest health centre** | | |
| Between 5–10 km | 225 | 7.1 |
| Below 5 km | 2966 | 92.9 |

Note: ANC: Antenatal care, km: kilometre

health facility, 99.7% were assisted by skilled health personnel, whereas health extension workers assisted 0.3% of deliveries.

Among mothers who had given birth at health facilities, 84.4% had normal, 11.2% had instrumental, and 4.4% had caesarean section deliveries. Mothers who had given birth at a health facility used an ambulance in 47% of cases, and 31.3% went by foot to reach the health

**Table 2. Delivery place and assistance during delivery, Sidama National Regional State, southern Ethiopia, 2020.**

| Variable | Number | Percent |
|---|---|---|
| **Delivery place** | | |
| Home | 1694 | 53.1 |
| Health post | 5 | 0.2 |
| Health centre | 931 | 29.2 |
| Hospital | 541 | 17.0 |
| Faith based clinic | 20 | 0.6 |
| **Mode of institutional delivery** | | |
| Normal | 1263 | 84.4 |
| Instrumental | 168 | 11.2 |
| Caesarean section | 66 | 4.4 |
| **Health facility delivery attendant** | | |
| Health extension worker | 5 | 0.3 |
| Nurse | 486 | 32.5 |
| Midwife | 803 | 53.6 |
| Doctor | 203 | 13.6 |
| **Attendant of home delivery** | | |
| Traditional birth attendant | 352 | 20.8 |
| Mother in law | 703 | 41.5 |
| Husband | 104 | 6.1 |
| Neighbour | 301 | 17.8 |
| No assistant/alone | 234 | 13.8 |
| **Reason for not giving birth at health facility** | | |
| Not necessary | 161 | 9.5 |
| Not customary | 982 | 58.0 |
| Preferred TBA | 89 | 5.3 |
| Health facility incur cost | 7 | 0.4 |
| Did not have money | 23 | 1.4 |
| Health facility far away | 178 | 10.5 |
| Transportation problem | 184 | 10.9 |
| Did not have accompany | 30 | 1.8 |
| Low quality service at health facility | 16 | 0.9 |
| Family did not allow | 24 | 1.4 |

Note: TBA: Traditional birth attendant

facility for delivery service. After reaching the first health facility, 12.8% of mothers were referred to another health facility.

Of mothers who had given birth at home, 20.8% were assisted by traditional birth attendants (TBAs), 41.5% by their mother-in-law, 6.1% by their husbands, and 17.8% by neighbours; 13.8% of home births had no assistant.

## Spatial accessibility, variations, and utilisation of skilled birth care

The mean distance from the women's residential locations to the nearest health centre was 2.6 km, ranging from 0 to 8.5 km. All 3191 women lived within 10 km of the nearest health centre.

Of the 1492 mothers assisted by skilled health personnel during birth, 1391 (93.2%) lived within 5 km and 101 (6.8%) between 5 and 10 km of the nearest health centre. Among the

**Table 3. Accessibility variations, Sidama National Regional State, southern Ethiopia, 2020.**

| District | Total women | Nearest Health Centre | | | | Nearest Hospital | | | | Referral hospital, Hawassa |
|---|---|---|---|---|---|---|---|---|---|---|
| | | Mean distance (km) | Maximum distance (km) | Women residing outside 5km | % | Mean distance (km) | Maximum distance (km) | Women residing outside 10km | % | Mean distance (km) from district to referral centre |
| Aleta Chuko | 596 | 3 | 8.5 | 124 | 20.8 | 6.7 | 12.4 | 55 | 9.2 | 61 |
| Aleta Wondo | 704 | 1.7 | 6.4 | 1 | 0.1 | 6.4 | 14.0 | 97 | 13.8 | 64 |
| Aroresa | 503 | 3.4 | 5.8 | 64 | 12.7 | 6.8 | 14.2 | 172 | 34.2 | 181 |
| Daela | 205 | 3.4 | 6.9 | 36 | 17.6 | 12.5 | 14.8 | 205 | 100.0 | 175 |
| Hawassa Zuriya | 658 | 2.5 | 4.9 | 0 | 0.0 | 5.4 | 18.6 | 1 | 0.2 | 21 |
| Wondogenet | 525 | 2.3 | 4.6 | 0 | 0.0 | 17.0 | 21.9 | 522 | 99.4 | 25 |
| Total | 3191 | 2.6 | 8.5 | 225 | 7.1 | 8.5 | 21.9 | 1052 | 33.0 | |

Note: Km: kilometre

1699 mothers who gave birth at home, 1551 (91.3%) lived within 5 km and 148 (8.7%) between 5 and 10 km of the nearest health centre.

The mean distance from the women's residential locations to the nearest hospital was 8.5 km, ranging from 1 to 22 km. Two-thirds of the women, 2137 (67%), resided within 10 km, whereas one-third, 1054 (33%), lived in areas more than 10 km from the nearest hospital.

Of the 1492 mothers who had given birth assisted by skilled health personnel, 1215 (81.4%) lived within 10 km, while 277 (19.6%) lived more than 10 km from the nearest hospital. Among the 1699 mothers who had given birth at home, 922 (54.3%) lived within 10 km, and 777 (45.7%) lived more than 10 km from the nearest hospital.

Table 3 shows household and district-level accessibility variations to the nearest health centre, the nearest hospital, and the regional referral centre, Hawassa. The proportion of women living more than 5 km from the nearest health centre was 20.8% in Aleta Chuko district, 17.6% in Daela, and 12.7% in Aroresa. Three districts had a high proportion of women living more than 10 km from the nearest hospital: Daela (100%), Wondogenet (99.4%), and Aroresa (34.2%). The same districts had the lowest SBA. Aroresa and Daela districts are considered remote districts located 181 km and 175 km from the regional centre, Hawassa, respectively.

## Variations in the use of skilled birth attendants and maternal mortality

Table 4 describes the proportion of deliveries with skilled birth attendants in districts of Sidama Region and the corresponding maternal mortality ratio (MMR). The coverage of SBA

**Table 4. Skilled birth attendants and maternal mortality ratio by district, Sidama National Regional State, southern Ethiopia, 2020.**

| District | Skilled birth attendant (SBA) | | | | Maternal mortality ratio (MMR) | |
|---|---|---|---|---|---|---|
| | Total mothers interviewed | Mothers used SBA | % SBA | 95% CI | MMR | 95% CI |
| Aleta Chuko | 596 | 375 | 62. 9 | 59.0–66.7 | 263 | 58–467 |
| Aleta Wondo | 704 | 498 | 70. 7 | 67.3–74.0 | 525 | 207–842 |
| Aroresa | 503 | 99 | 19. 7 | 16.4–23.4 | 1142 | 693–1591 |
| Daela | 205 | 11 | 5. 4 | 3.0–9.4 | 641 | 77–1358 |
| Hawassa Zuriya | 658 | 368 | 55. 9 | 52.1–59.7 | 114 | 24–251 |
| Wondogenet | 525 | 141 | 26. 8 | 23.2–30.8 | 258 | 8–508 |

Note: CI: Confidence interval

was less than 50% in three out of six districts. Daela district had the lowest SBA coverage; 5.4% (95% CI: 3.0–9.4), followed by Aroresa; 19.7% (95% CI: 16.4–23.4), and Wondogenet at 26.8% (95% CI: 23.2–30.8). The two districts with the lowest coverage of SBA, Aroresa and Daela districts, had the highest MMR, with Aroresa district having a MMR of 1142 (95% CI: 693–1591) per 100,000 LB and Daela district having a MMR of 641 (95% CI: 77–1358) per 100,000 LB.

## Factors associated with skilled birth attendants

Table 5 shows the result of the mixed-effect multilevel logistic regression (MLLR) analysis. The likelihood of utilization of SBA was higher among mothers with formal education (AOR: 1.5; 95% CI: 1.1–2.0), whose husbands were not engaged in farming (AOR: 1.5; 95% CI: 1.2–1.9), who experienced pregnancy-related complications (AOR: 1.7; 95% CI: 1.4–2.2), and who attended ANC visits (AOR: 10.3; 95% CI: 7.5–14.1). Multiparous women were less likely to use skilled birth (AOR: 0.5; 95% CI: 0.4–0.6).

The intra-cluster correlation (ICC) for the districts was 0.30 (95% CI: 0.24–0.36), meaning that 30% of the variations in use of SBA were attributable to the district-level differences. We did not find significant ICC for *limatbudin* and kebele levels.

**Table 5. Multilevel logistic regression analysis of skilled birth attendants, Sidama National Regional State, southern Ethiopia, 2020.**

| Variable | | Type of birth | | Bivariate analysis | Multilevel Logistic Regression Analysis | |
|---|---|---|---|---|---|---|
| Fixed effect | Total | Skilled | Unskilled | COR (95% CI) | AOR (95% CI) | P-value |
| **Age of mother** | 3191 | | | 0.9 (0.8–0.9) | 0.9 (0.9–1.0) | 0.252 |
| **Education of mother** | | | | | | |
| No formal education | 698 | 184 | 514 | 1 | 1 | |
| Formal education | 2493 | 1308 | 1185 | 3.1 (2.6–3.7) | 1.5 (1.1–2.0) | 0.041 |
| **Education of husband** | | | | | | |
| No formal education | 420 | 120 | 300 | 1 | 1 | |
| Formal education | 2771 | 1372 | 1399 | 2.4 (1.9–3.1) | 1.0 (0.7–1.4) | 0.817 |
| **Wealth index of household** | | | | | | |
| Poor | 1932 | 933 | 999 | 1 | 1 | |
| Rich | 1259 | 559 | 700 | 0.8 (0.7–1.0) | 1.0 (0.8–1.3) | 0.548 |
| **Occupation of husband** | | | | | | |
| Farming | 2276 | 966 | 1310 | 1 | 1 | |
| Non-farming* | 915 | 526 | 389 | 1.8 (1.6–2.1) | 1.5 (1.2–1.9) | <0.001 |
| **Parity** | | | | | | |
| Nulipara | 1063 | 624 | 439 | 1 | 1 | |
| Multipara | 2128 | 868 | 1260 | 0.5 (0.4–0.6) | 0.5 (0.4–0.6) | <0.001 |
| **ANC** | | | | | | |
| No | 790 | 86 | 704 | 1 | 1 | |
| Yes | 2401 | 1406 | 995 | 11.6 (9.1–14.7) | 10.3 (7.5–14.1) | <0.001 |
| **Pregnancy related complication** | | | | | | |
| Had no complication | 1945 | 838 | 1107 | 1 | 1 | |
| Had complication/s | 1246 | 654 | 592 | 1.4 (1.3–1.7) | 1.7 (1.4–2.2) | <0.001 |
| **Distance (nearest Hosp. in km)** | 3191 | | | 0.89 (0.87–0.90) | 0.86 (0.83–0.89) | 0.001 |
| **Distance (nearest HC in km)** | 3191 | | | 0.76 (0.72–0.80) | 0.75 (0.67–0.85) | <0.001 |
| **Random effect** | | | | | | |
| ICC District | | | | | 0.30 (0.24–0.36) | |
| AIC | | | | | 3088 | |

Note: *Trading, government employee etc. AIC: Akaike's Information Criterion, AOR: Adjusted odds ratio CI: Confidence interval, COR: Crude odds ratio, ICC: Intra-cluster correlation, HC: Health centre, Hosp: Hospital

Distance to the nearest facility was associated with using SBA: a 1 km increase in distance to the hospital decreased the proportion of SBA by 14%; a 1km increase in distance to the health centre decreased the proportion of SBA deliveries by 25%.

## Discussion

In the Sidama region, the proportion of births attended by skilled health personnel was 46.7%, with large variations between the districts. Thirty per cent of variations in SBA are accounted for by the differences among the districts. Maternal mortality was high in districts where the proportion of births attended by skilled personnel was low. Factors associated with the use of SBA were the mother's education, the husband's occupation, pregnancy-related complications, ANC, parity, and distance to the nearest hospital and health centre.

In this study, we used the geographic coordinates of the nearest health centres and hospitals and coordinates collected at the individual household level to estimate the accessibility of the nearest health centres and hospitals as opposed to the commonly used village centroid.

This study had some limitations. Recall bias is a limitation of the study, as we asked about women's birth experiences in the past two years. However, giving birth is memorable, and recall bias is less likely. Social desirability bias: participants may state what they think is the acceptable response in the community, which can influence our results. However, we used different verification and probing questions to confirm the SBA. We could not establish a temporal relationship between the outcome and exposure variables due to the study's cross-sectional nature.

We could not find data on roads and modern transportation networks as the study was done in a rural area. Hence, we used Euclidean (straight line) distance to measure the geographic accessibility of health centres and hospitals for SBA. The Euclidean method is recommended in rural settings to measure the accessibility of health facilities [35, 36]. Studies have also documented that in a rural setting, estimates from straight line distance measurements are comparable with the results of driving distance and driving time [35].

We may have missed births for women who are unmarried. However, we believe that in a rural setting, the majority of pregnancies are a result of marriage. Studies have shown that unmarried women do not constitute a significant number in such studies [12, 14]. We did not also find women who had a history of pregnancy below the age of 15 years.

Since the crude and adjusted risk factors vary substantially (Table 5), we carried out stratified analysis to identify and control for confounding [40]. When we analysed the association between husband education and SBA, stratified by mother education, the difference between crude and adjusted odds ratios remained markedly different. This indicates that mother education is a confounder of husband education, as it is associated with the exposure (husband education) and the outcome (SBA). In other words, an educated husband is more likely to marry an educated woman, and mother education is a risk factor for SBA. In addition, there might be residual confounders that we did not consider in our study, which may have affected our results.

In the Sidama Region, the proportion of deliveries using SBA was 46.7%, with large variations in the districts. This finding is comparable with the findings of other studies in the country [16, 24, 37]. However, it is below the national plan 2020, and the target set for 2025 by Health Sector Transformation Plan II [41]. The global proportion of births using SBA is 84%; however, in this study, the coverage of SBA is below the level of the sub-Saharan African countries, at 52% [4]. The very low coverage of SBA in some districts in Sidama is attributable to the inaccessibility of health facilities, difficult topography, lack of transportation, poor referral systems, and a shortage of skilled health professionals. Urgent action and concerted effort are

needed to increase the use of SBA in the Sidama region and to support the country's efforts towards attaining the Sustainable Development Goal.

This study found that around one-third of the variation in SBA was attributed to district differences. This result is higher than the findings of a national study conducted in Ethiopia, which found around 22% of the variation in SBA was attributed to differences between communities [42]. The differences between the two studies can be explained by the fact that in our study, the ICC of the districts was considered, whereas the national study used enumeration locations to estimate the ICC. The high variations in SBA attributable to the differences between the districts in our study could be explained by the uneven distribution of health services, including the differences in the mix and number of skilled health professionals, the availability of the necessary logistics and supplies for maternal health services, and other infrastructure like roads, water supply, and electricity in the districts.

This study shows that districts with a low proportion of deliveries attended by SBA have a high maternal mortality. Similar findings have been reported in the 2022 UNICEF report [4]. In places with low SBA, many mothers give birth at home, unattended by skilled health professionals and without access to life-saving drugs and interventions. This leads to the loss of many mothers and new-borns due to the risks associated with unattended home delivery [43]. Improving access, quality, and coverage of key interventions, including SBA, saves the lives of mothers and new-borns [44].

Wondogenet district is an exception, with a fairly low MMR despite a low proportion of deliveries using SBA. The probable reason for lower MMR is that the district has better roads, available transportation for emergencies, a functional local referral system, and fairly easy access to the regional referral centre in Hawassa.

The use of SBA was more common among mothers with an education. This finding agrees with the studies done in Ethiopia [12, 25] and elsewhere [9, 45]. Mothers' formal education is associated with better health knowledge and, consequently, with improved use of health services [46]. Educated women have higher healthcare decision autonomy [46] and are more likely to plan and prepare for an obstetric emergency [47]; education empowers women, and consequently, more of them use SBA [48]. However, the association between mother education and SBA is weak in our study. This shows other factors like ANC, pregnancy complications, distance to the nearest hospital, and distribution of services within the districts play a great role in using SBA.

The probability of using SBA increased for women who had complications during pregnancy. Studies conducted in northwest Ethiopia [12] and Bangladesh [49] reported similar findings. This association can be explained by the fact that mothers with pregnancy complications are more likely to seek medical care, be advised on the importance of SBA during ANC, and be referred for SBA [50].

This study demonstrated that attendance at antenatal care is a strong predictor of using SBA. This finding agrees with the studies conducted in Kenya [51] and Ethiopia [25]. This could be explained by the fact that women who attend ANC will be familiar with the health system and health facilities, be encouraged and counselled on the importance of SBA by ANC providers, and prepare for facility delivery.

Compared to primiparous women, multiparous women were less likely to use SBA. A similar finding has been reported by a study conducted in southern Ghana and Tanzania [19, 52]. Multiparous women may be influenced by their previous experiences and consider childbirth natural. Moreover, multiparous women may have strong cultural ties and be more influenced by traditional practices.

The use of SBA during deliveries is more common among women married to employed men or traders. A similar finding has been reported from a study conducted in Nigeria [53].

This could be explained by the fact that employed husbands are more likely to be educated, better understand the use of SBA, and support their wives to take SBA care and have a better income, making it easier for the family to cover the costs associated with skilled delivery care.

According to the WHO, pregnant women should be able to access EmONC health facilities within 2 hours of travel time [38, 54]. Our study found that 33% of women reside more than 10 km from the nearest hospital, with great variations by district, meaning a third of the women must travel more than 2 hours to access the nearest hospital. Districts where the women live far from the nearest hospital had fewer women assisted by SBA for their deliveries. It has also been recommended that, at the national or subnational level, at least 80% of pregnant women should access the nearest hospital within 2 hours of travel time [54, 55], but we found in our study three districts where the proportion of pregnant women who access the nearest hospital in 2 hours is below the recommended 80%.

The low proportion of births attended by SBA with high variations in districts shows the Sidama National Regional Health Bureau and district health offices should take actions to improve the use of SBA, identify risk factors, and design interventions tailored to the local context. The association of low SBA with high maternal mortality in districts far away from the regional centre reaffirms that SBA plays a key role in reducing maternal mortality. In order to improve SBA coverage and decrease maternal mortality, the government and other concerned bodies are recommended to give due attention to alleviating the uneven distribution of maternal health services, a lack of skilled health personnel, and poor infrastructure in districts located in remote and distant areas. The construction of hospitals in areas where women face difficulty accessing hospital services will likely improve the use of SBA. Encouraging all pregnant women to get ANC services may also improve the use of SBA services.

## Conclusions

In the Sidama Region, the coverage of skilled birth attendant is generally low but varies greatly by districts. Low-skilled birth attendant was associated with high maternal mortality. Corrective actions are needed to improve the low coverage of skilled birth attendants and decrease the high maternal mortality. The accessibility of hospitals to pregnant and labouring women has to improve. Increasing the use of antenatal care by pregnant women will improve the coverage of skilled birth attendants.

## Supporting information

**S1 Checklist. Inclusivity in global research.**
(DOCX)

## Acknowledgments

We want to thank the participants in the study for their time and information. Our special thanks go to the Sidama National Regional State Health Bureau, respective district health offices, and kebele administrations in Ethiopia for their support and permission to conduct the study. We thank data collectors and clerks for their time and commitment.

## Author Contributions

**Conceptualization:** Aschenaki Zerihun Kea, Bernt Lindtjørn, Sven Gudmund Hinderaker.

**Data curation:** Aschenaki Zerihun Kea.

**Formal analysis:** Aschenaki Zerihun Kea.

**Investigation:** Aschenaki Zerihun Kea, Bernt Lindtjørn, Sven Gudmund Hinderaker.

**Methodology:** Aschenaki Zerihun Kea, Bernt Lindtjørn, Sven Gudmund Hinderaker.

**Project administration:** Aschenaki Zerihun Kea.

**Supervision:** Bernt Lindtjørn, Achamyelesh Gebretsadik Tekle, Sven Gudmund Hinderaker.

**Writing – original draft:** Aschenaki Zerihun Kea.

**Writing – review & editing:** Aschenaki Zerihun Kea, Bernt Lindtjørn, Achamyelesh Gebretsadik Tekle, Sven Gudmund Hinderaker.

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
