## [Decision Letter · Decision Letter 0]

16 Oct 2023

PGPH-D-23-01804

Southern Ethiopian skilled birth attendant variations and maternal mortality: a multilevel study of a population-based cross-sectional household survey

Dear Kea,

Thank you for submitting your manuscript to PLOS Global Public Health. After careful consideration, we feel that it has merit but does not fully meet PLOS Global Public Health’s publication criteria as it currently stands. Therefore, we invite you to submit a revised version of the manuscript that addresses the points raised during the review process.

We look forward to receiving your revised manuscript.

Kind regards,

Collins Otieno Asweto, PhD

Academic Editor

Journal Requirements:

Additional Editor Comments (if provided):

Reviewers' comments:

Reviewer's Responses to Questions

**Comments to the Author**

1. Does this manuscript meet PLOS Global Public Health’s publication criteria? Is the manuscript technically sound, and do the data support the conclusions? The manuscript must describe methodologically and ethically rigorous research with conclusions that are appropriately drawn based on the data presented.

Reviewer #1: Yes

Reviewer #2: Yes

2. Has the statistical analysis been performed appropriately and rigorously?

Reviewer #1: Yes

Reviewer #2: Yes

3. Have the authors made all data underlying the findings in their manuscript fully available (please refer to the Data Availability Statement at the start of the manuscript PDF file)?

Reviewer #1: No

Reviewer #2: No

4. Is the manuscript presented in an intelligible fashion and written in standard English?

Reviewer #1: Yes

Reviewer #2: Yes

5. Review Comments to the Author

Reviewer #1: First of all, this is a very important study addressing the importance and consequences of using skilled birth attendants (SBAs) in safeguarding safe motherhood principles. The findings of the study have potentials to impact public health policy and programme development. I have made a number of observations with regards to the manuscript.

1. The objectives of the study are many and important but need to presented more clearly.

2. The methodology comprising the study population, the sampling design, data analysis, and ethical consideration are comprehensively presented in the manuscript.

3. The section in the methods that I think the authors need to address is the sample size presented. In the abstract, 3191 women were said to have been interviewed but in the sample determination, figures such as 768 and 8880 are also talked about. There appears to be contradiction here and so, please, reconcile and be consistent.

4. In the introduction, the authors seem to be reporting results of their study in lines 84-91. Please, check and reserve your findings to the results section.

5. The policy recommendations appear to sound like authorizing or commanding instead of advocacy. Please, check.

Despite the above, the study is very comprehensive and well conducted on a very critical maternal, child, and reproductive health issue. Please, check my comments in the PDF attached.

Reviewer #2: Overall, Aschenaki et al. discuss the coverage of SBA in Sidama national regional state and its correlation with maternal mortality. This topic is very relevant and of great public health importance in Sub-Saharan Africa. Overall, the article is well written and the information is clearly presented. The conclusion are in line with the research findings.

Abstract

The abstract is well written. It summarises the important aspects of the study well. However, to comply with the journal guidelines, subheadings may be eliminated.

Introduction.

The introduction gives a good preamble to the research topic. It sets a good background for the reader to understand this research topic. However, it would be nice to describe the setting of Sidama national region including the physical, social, economic structures as this helps the reader to understand the research context better.

Methods

The methods are described sufficiently for the research to be reproduced.

Results

The results are well presented. The effect of larger sampling units on the results was accounted for, which is good. However, the number of women interviewed is a lot higher than the proposed sample size, is there an explanation for this?

The data in support of these has not been provided yet and therefore, the results cannot be proven.

6. PLOS authors have the option to publish the peer review history of their article (what does this mean?). If published, this will include your full peer review and any attached files.

**Do you want your identity to be public for this peer review?** For information about this choice, including consent withdrawal, please see our Privacy Policy.

Reviewer #1: No

Reviewer #2: No

---

## [Decision Letter · Decision Letter 1]

27 Nov 2023

Southern Ethiopian skilled birth attendant variations and maternal mortality: a multilevel study of a population-based cross-sectional household survey

PGPH-D-23-01804R1

Dear Kea,

We are pleased to inform you that your manuscript 'Southern Ethiopian skilled birth attendant variations and maternal mortality: a multilevel study of a population-based cross-sectional household survey' has been provisionally accepted for publication in PLOS Global Public Health.

Best regards,

Collins Otieno Asweto, PhD

Academic Editor

Reviewer Comments

Take note on minor changes as recommended by Reviewer 3:

Reviewer's Responses to Questions

**Comments to the Author**

1. If the authors have adequately addressed your comments raised in a previous round of review and you feel that this manuscript is now acceptable for publication, you may indicate that here to bypass the “Comments to the Author” section, enter your conflict of interest statement in the “Confidential to Editor” section, and submit your "Accept" recommendation.

Reviewer #1: All comments have been addressed

Reviewer #2: All comments have been addressed

Reviewer #3: (No Response)

2. Does this manuscript meet PLOS Global Public Health’s publication criteria? Is the manuscript technically sound, and do the data support the conclusions? The manuscript must describe methodologically and ethically rigorous research with conclusions that are appropriately drawn based on the data presented.

Reviewer #1: Yes

Reviewer #2: Yes

Reviewer #3: Yes

3. Has the statistical analysis been performed appropriately and rigorously?

Reviewer #1: Yes

Reviewer #2: Yes

Reviewer #3: Yes

4. Have the authors made all data underlying the findings in their manuscript fully available (please refer to the Data Availability Statement at the start of the manuscript PDF file)?

Reviewer #1: Yes

Reviewer #2: No

Reviewer #3: Yes

5. Is the manuscript presented in an intelligible fashion and written in standard English?

Reviewer #1: Yes

Reviewer #2: Yes

Reviewer #3: Yes

6. Review Comments to the Author

Reviewer #1: I have reviewed the entire manuscript a number of times and have observed that all my concerns and recommendations have been adequately addressed. The manuscript has, therefore, become much clearer and meaningful to the reader.

Reviewer #2: All previous comments addressed. no additional comments.

Reviewer #3: I am reviewing a revised version of this manuscript, specifically the Revised Manuscript with Track Changes and am using the line numbering from that version. The authors have responded appropriately to the previous reviewers' comments and have made the necessary revisions in the manuscript.

This paper presents an analysis of the impact of skilled birth attendants on maternal mortality. The findings are important for decision making and resource allocation by the Ethiopian government and are relevant to many other LMICs. I feel that the paper is ready for publication, but would benefit from a small number of minor revisions to improve clarity and readability. Again, I am using the Line Numbering fro the "Revised Manuscript with Track Changes."

Line 67: Change "mother-in-law" to "mothers-in-law"

Line 75: change "...like.." to "...such as the level of husband education.."

Line 83: Replace the "and" with just a comma

Line 147: Again replace "mother-in-law" with "mothers-in-law"

Line 158: Change sentence to read, "...household characteristics such as the level of education of the husband,...."

Line 284: In this table change the heading of "Number of ANC visit" to Number of ANC visits"

Line 426, I would recommend that you change the sentence to read, "The probable reason for lower MMR is that the district has better roads."

Line 463: I would suggest that you change the sentence to read, "...women live far from the nearest hospital had fewer women assisted by SBA for their deliveries."

Line 469: I would suggest retaining the word, "should" rather than deleting it. This does represent an important recommendation that derives from the research described in this paper.

7. PLOS authors have the option to publish the peer review history of their article (what does this mean?). If published, this will include your full peer review and any attached files.

**Do you want your identity to be public for this peer review?** For information about this choice, including consent withdrawal, please see our Privacy Policy.

Reviewer #1: No

Reviewer #2: No

Reviewer #3: **Yes: **Paul R DeLay MD, DTM&H (Lond)
